# Mental Health and Homelessness in the Community of Madrid (Spain): The Impact of Discrimination and Violence

**DOI:** 10.3390/ijerph20032034

**Published:** 2023-01-22

**Authors:** Iria Noa de la Fuente-Roldán, Ana Isabel Corchado-Castillo, Ana Dorado-Barbé

**Affiliations:** 1Department of Social Work and Social Services (Faculty of Social Work), Complutense University of Madrid, 28223 Madrid, Spain; 2Institute for Research in Development and Cooperation (IUDC-UCM), Complutense University of Madrid, 28015 Madrid, Spain

**Keywords:** discrimination, homelessness, mental health, residential exclusion, violence

## Abstract

The aim of this study was to analyze the impact of experiences of violence and discrimination on mental health among people in situations of homelessness (PSH). For this purpose, a quantitative, descriptive, and correlational investigation was conducted by conducting a survey with 603 PSH living in the Community of Madrid (Spain). The results show high levels of mental health impairment, as well as approximately half of the participants having experienced discrimination and violence in the course of their homelessness. Perceived experiences of discrimination are associated with higher levels of mental health impairment (OR = 0.458; *p* = < 0.001; 95% IC 0.31–0.68). This deterioration is also related to a negative self-assessment of the general state of health among participants (OR = 0.262; *p* = < 0.001; 95% IC 0.12–0.57). However, impaired mental health is not associated with experiences of violence. The findings also indicate that there are intersections in terms of being female, young, and foreign that result in greater psychological impairment and a higher risk of experiencing violence and discrimination. This study provides an insight into the PSH experiences in relation to mental health, violence, and discrimination and the need to implement actions aimed at improving their psychosocial wellbeing from the perspective of respect for citizens’ rights.

## 1. Introduction

The processes of social change that have affected most European countries over the last 15 years (the 2008 crisis and the COVID-19 pandemic) have been particularly severe in Spain. According to Eurostat [1], 27.8% of the Spanish population was at risk of poverty and social exclusion in 2021. This figure is almost seven percentage points higher than the European average (EU-28; 21.2%) and over four percentage points higher than the Spanish rate in 2007 (23.3%) [1]. This trend for increasing social exclusion in Spain is ongoing, with a gap of 2.5% compared to 2019 (25.3%), producing a panorama with notable social inequalities in one of the most highly developed countries. These social inequalities give rise to different realities, with homelessness being one of their most visible and extreme manifestations.

Homelessness is a social, historical, and cultural reality [2] affecting at least 28,552 citizens in Spain [3]. The population of people in situations of homelessness (PSH) in the Community of Madrid is estimated at 4146 people [3], of whom 83.8% are men and 16.2% are women. This proportion differs from the national-level data, where women represent 23.3% of the PSH. In the Community of Madrid, 44.6% of PSH are Spanish-born and 55.4% are of foreign origin. In terms of age, 13.6% of PSH living in the Community of Madrid are below 30 years old, while 25.7% are aged between 30 and 44, 54.7% are between 45 and 64 and 6% are over 65 years old [3].

Homelessness in Spain has been extensively analyzed and studied since the 1990s, with research notably focusing on analyzing the homeless population and the different dimensions that shape their social exclusion [4,5,6,7,8,9]. This research has shown homelessness to be a complex issue that impacts the most vulnerable social groups, necessitating a comprehensive understanding of the various factors that impact the lives of those it affects [10]. The literature examining homelessness places particular emphasis on analyzing the factors that contribute to its appearance and persistence [11,12,13].

In this regard, Fitzpatrick [14] distinguishes three levels of analysis of homelessness. The first, macro-level analyses are related to the labor market, employment, and housing. The second, meso-level incorporates the role played by interpersonal relationships (including social support and social networks). The final, micro level focuses on individual aspects (including health, age, and disability). Notable studies have analyzed dimensions including social and housing policies and situations of crisis [15], social support and social relationships [16], quality of life [12], the consumption of alcohol [13] and other addictive substances [17], and, most importantly, aspects related to physical and mental health [11,18].

### 1.1. Homeless and Mental Health

A range of studies have examined the impact of homelessness on health and psychological wellbeing. These studies have reported the impact that being homeless has on people, causing them to face a lower life expectancy than the general population [19], a higher risk of death [20], and an increased prevalence of illness [21], including mental illness [22,23].

The prevalence and presence of mental illness and the role played by psychological and physical problems in the appearance and persistence of homelessness is one of the most examined aspects in the literature [24]. At the same time, mental health problems have been identified as a specific pathway to homelessness [25] that is estimated to affect between 25% and 35% of PSH [26]. 

There is a relatively extensive line of research examining the impact and scale of mental health problems affecting PSH. Studies have notably examined life events that act as stressors and are more prevalent among PSH than among the general population [11,27], such as depressive symptoms and other mental illnesses [28,29] and suicide attempts [30]. 

However, the scale or prevalence of mental health problems is not the only thing that requires their examination, which can also act as a starting point in investigating the specific impact that mental health problems have on the realities and day-to-day lives of PSH. Previous research and experience in psychosocial intervention show that PSH face multiple stressors that contribute to increasing the risk of the deterioration of their mental health [26,31]. This means that mental illness can drive extreme social exclusion but can also develop as a consequence of the numerous stressors that affect the life pathways of people in situations of homelessness (PSH). As reported by Jafry et al. [32], these stressors notably include exposure to violence and discrimination. The recognition of vulnerability linked to situations of homelessness [33] has caused an increasing emphasis in recent years on specifically examining the impact of the discrimination and violence that affect these citizens [34].

### 1.2. Homelessness, Discrimination, and Violence 

Various studies have reported that PSH face greater situations of risk [34] and have more experiences of victimization compared to the population that is not affected by processes of social exclusion [35].

In this regard, increasing importance has been placed in Spain on the study of aporophobia, a specific form of discrimination that, as defined by Cortina [36], affects poor people merely because they are in a situation of poverty. This is so significant that aporophobia has been added to the Spanish Criminal Code (article 22.4) as an aggravating circumstance for criminal offences.

In Spain, around half of PSH have felt discriminated against due to their homelessness [3]. 

In addition, the Observatory for Hate Crimes against People in Situations of Homelessness (HATENTO) [37] reports that almost half of PSH have been victims of violence due to being in a situation of homelessness. Specifically, 50.3% of PSH in Spain state that they have suffered from some form of criminal offence or assault since becoming homeless [3]. 

Calvo et al. [33] has found that 76.2% of PSH in Spain have been the victims of at least one violent episode in the previous year. Along the same lines, Muñoz et al. [38] found that 55.1% of people sleeping on the street had been victims of assault since becoming affected by homelessness. These studies, albeit with minor variations, show that violence is a reality in the lives of these citizens, meaning that PSH face more experiences of victimization than the population that is not affected by processes of social exclusion [35,39].

These experiences are particularly important to consider due to their impact on the lives of PSH. The consequences are both physical and psychological [40,41]. As reported by Meinbresse et al. [42], experiences of violence and discrimination increase the risk of suffering from psychological impairment or exacerbate an already precarious state of mental health.

### 1.3. The Present Propossal

As seen above, there is a trend for research that focuses on the scope and characterization of mental health among PSH, as well as their experiences of violence and discrimination. However, there are some limitations in this respect.

Mental health has been extensively examined, but there is a difficulty due to the absence of epidemiological studies that would make it possible to analyze the true scale of mental health problems, as well as their role in the appearance and persistence of homelessness [26]. This means that there is a need for a continuously updated examination of the mental health difficulties affecting this population.

In addition, the violence and discrimination suffered by PSH are not yet a central element in the analysis of homelessness, despite its scale. This issue has not been specifically addressed. This is especially pertinent if one takes into account the impact that these experiences can have on PSH and, specifically, on the definition of their state of mental health. Mental health, violence, and discrimination are commonly addressed in the context of homelessness but have traditionally been the object of independent analyses, without investigating their potential relationship.

Finally, a review of the literature shows that the realities of mental health and experiences of violence and discrimination are not the same for everyone [40]. Although some studies report that women [43], young people [3], people living on the street or in shelters [25], and immigrants [26] have more impaired mental health statuses and face a higher risk of violence and discrimination, intersectional approaches remain scarce. Differences associated with age, gender, nationality, and housing situation, as well as their intersectional impact, are considered to play a fundamental role in this regard [24]. Intersectionality comprises a specific way of examining social inequalities. It can be affirmed that the impact of violence and discrimination on mental health depends on the individual’s position in relation to a series of inequality factors. As stated above, these factors notably include gender, age, and nationality [44].

This differentiated impact of mental health, violence, and discrimination in the context of homelessness creates a need for research that examines its scope, dimensions, specific aspects, and consequences. In this regard, the overall aim of this study was to analyze the impact that experiences of discrimination and violence have on mental health among PSH. Based on the observations made, and in the interest of exploring the limitations encountered, five hypotheses were proposed: (1) PSH have high levels of mental health impairment; (2) mental health impairment among PSH is related to the sociodemographic variables of age, sex, nationality, and housing situation; (3) homelessness creates situations of risk associated with violence and discrimination; (4) experiences of perceived discrimination among PSH are associated with mental health impairment; (5) experiences of violence among PSH are linked to mental health impairment.

## 2. Materials and Methods

The transversal, descriptive, and correlational study was conducted using a quantitative methodological design. The study was applied by means of a survey. 

### 2.1. Sample and Participants

The population of PSH in the Community of Madrid is 4.146 people [3]. A sample of 573 PSH was calculated, with a margin of error of 5% and a confidence level of 99%. Finally, the sample was made up of 603 PSH living in the Community of Madrid (Spain). Convenience sampling was carried out based on three theoretical criteria (quotas). The first criterion consisted of an operational definition of homelessness based on the European Typology of Homelessness and Housing Exclusion (ETHOS) [45], which distinguishes four conceptual categories: rooflessness (ETHOS 1), houselessness (ETHOS 2), living in insecure housing (ETHOS 3), and living in inadequate housing (ETHOS 4). These categories have been grouped into two larger concepts: literal homelessness (living on the street or in centers for PSH: ETHOS 1 and 2) and broad homelessness (precarious housing, housed out of necessity, forced cohabitation, etc.: ETHOS 3 and 4) [46]. This approach makes it possible to broaden the view of homelessness beyond its most visible forms [47]. Quotas were also established by sex (male/female) and nationality (foreign/non-foreign).

### 2.2. Procedure

The questionnaire was adapted into an online version (LimeSurvey), and it was administered by research team members between February and May 2021 via personal face-to-face interviews conducted at various social care centers in the Community of Madrid (drop-in centers, shelters, emergency centers, and food kitchens). The researchers asked the questions and used LimeSurvey to record the answers given by participants. The in-person administration of the survey meant that it was possible to provide explanations where required for reasons such as language barriers or other communication difficulties.

The participants were not compensated for their participation. They were asked to voluntarily take part in the research and were assured that their responses would be confidential. 

The interviews conducted to administer the survey took an average of 60 min. Professionals at the various centers provided support to recruit participants. The participants signed an informed consent form before taking part in the study. The study was approved by the Research Ethics Committee of Complutense University, Madrid (Ref: CE_20210415-02_SOC). 

### 2.3. Variables and Measures

Outcome variable. The 12-item version of the General Health Questionnaire (GHQ-12) [48], adapted into and validated for Spanish by Rocha et al. [49], was used to examine mental health. The purpose of this screening instrument is to detect psychological morbidity and general mental health problems among the general population [50]. It is made up of 12 items that are answered using a Likert-type scale ranging from 0 to 3 for each response option. Following Rocha et al. [49], scores equal to or higher than 3 meet the criterion for a potential case of impaired mental health. It is frequently used to analyze the mental health of homelessness [31,51,52].

An exploratory factor analysis was conducted to assess the unidimensionality of the GHQ-12 [53]. The parallel analysis suggested a single factor with good statistical values: KMO = 0.85 and UniCo = 0.95. GHQ-12 shows adequate reliability, with McDonald’s ordinal Omega = 0.86 and Standardized Cronbach’s Alpha = 0.86, revealing high internal consistency.

Exposure variables. The questions included in the Spanish Survey on Homelessness were used to measure the exposure variables (perceived discrimination and types of violence suffered) [3]:Perceived discrimination. PSH were asked if they had ever felt discriminated against due to being in a situation of homelessness. The following response options were included: “yes, I have felt discriminated against”; “no, I have not felt discriminated against”; and “don’t know/no response”.Types of violence suffered. PSH were asked if they had been the victim of any criminal offence while in a situation of homelessness. The following response categories were offered: “No, I have not suffered violence”; “I have suffered physical violence”; “I have suffered some form of sexual assault”; “I have suffered verbal assaults (insults and/or threats); and “don’t know/no response”. This question was recoded into three response categories: “No, I have not suffered violence”; “I have suffered violence” and “Don’t know/no response”. The aim was to obtain data on people who have suffered from violence and people who have not suffered from violence.

Control variables. The study includes self-rated health and sociodemographic variables as control variables:Self-rated health. This is a significant variable insofar as it affects living conditions and health, including mental health, among PSH [54]. This variable was examined using a single question: “how would you describe your current state of health?” Five response options were offered, ranging from “very poor” (1) to “very good” (5). The variable was dichotomized into good self-perceived health (very good and good) and poor self-perceived health (average, poor, and very poor). This is a widely used way of asking about self-perceived general states of health [55].Sociodemographic variables. Variables that are commonly used with PSH were included [24]: sex (male/female); nationality (Spanish/Latin American/African/European); age (quantitative variable, recodified into the following ranges: 35 years or younger; 36–50 years; 51 years or older); and housing situation (literally homeless/broadly homeless).

### 2.4. Analysis

A univariate and descriptive analysis of the study variables was carried out first. Second, the Chi-Squared statistic was used to analyze the relationship between the study variables (outcome variable, exposure variables, and control variables). Cramér’s V and Phi were also used to identify the size of the effect of the analysis. Third, a logistic regression analysis was performed based on four theoretical models to examine the relationship findings in depth. The first model incorporated the sociodemographic variables (sex, age, nationality, and housing situation). Model 2 incorporated self-assessed health. Model 3 introduced perceived discrimination, and the final model added the variables related to the types of violence suffered by the participants. The data analysis was conducted using the IBM-SPSS program (v.28). The FACTOR program (v.12) was used for the exploratory factor analysis and the reliability statistics of GHQ-12, using McDonald’s omega and ordinal Cronbach’s Alpha.

## 3. Results

### 3.1. Descriptive and Bivariate Analyses

As set out in Table 1, the most common participant profile was male, aged over 51 years, Spanish and in a literally homeless situation. The average sample age was 46.29 years (SD = 14.76; min 19–max 80). Specifically, 28.9% of the sample were aged 35 years or younger, 24.4% were between 36 and 50 years old, and 46.6% were 51 or older. The sample comprised more men (65.5%) than women (34%). In terms of nationality, 37.6% of the participants were Spanish and 62.4% were foreign (with 9.5% coming from another European country, 21.6% coming from Africa, and 31.3% coming from Latin America). Finally, in terms of housing situation, 46.8% of the participants were literally homeless and 52.7% were broadly homeless.

Table 2 sets out the summary of the descriptive results for the main study variables. There was a notably high average score for GHQ-12 (average = 5.03; SD = 3.41), indicating high levels of psychological impairment. Taking into account the cut-off point that was used (equal to or greater than 3) [28], 66.9% of the study participants reported impaired mental health. On the other hand, a majority of the participants gave a positive assessment of their general state of health (87.4%).

Perceived discrimination due to homelessness was reported by 48.4% of the sample, while 44.8% of the participants stated that they had experienced violence. Specifically, 26.4% stated that they had suffered from verbal assaults, 34.5% reported having been subjected to physical assaults, and 4.6% of the sample reported having suffered from sexual assaults.

With regard to the relationships between impaired mental health and the other study variables (Table 3), the findings showed a statistically significant relationship between impaired mental health and the variables of sex (χ^2^ (1) = 13.18; Phi = 0.15; *p* < 0.001), nationality (χ^2^ (3) = 12.88; Phi = 0.15; *p* = 0.005), self-assessed health (χ^2^ (1) = 14.23; Phi = 0.16; *p* < 0.001), perceived discrimination (χ^2^ (1) = 16.03; Phi = 0.16; *p* = < 0.001), and age (χ^2^ (2) = 11.53; Phi = 0.14; *p* = 0.003). Neither housing situation nor the different types of violence (physical, sexual, and verbal) presented statistically significant differences in relation to mental health.

Specifically for age, assuming an equality of variances (Levene test: F = 0.04; *p* = 0.83), the participants with impaired mental health were significantly younger (M = 49.87, SD = 14.94) than those without impaired mental health (M = 44.57; SD = 14.67; t (600) = 4.00; *p* < 0.001; d = 0.36). In addition, and as set out in Table 3, women, people who gave a negative self-assessment of their health, people of Latin American origin, and participants who perceived that they had suffered from discrimination all reported higher levels of impaired mental health. 

In terms of the relationships between perceived discrimination and the other study variables, the findings showed a significant relationship between having experienced perceived discrimination and self-assessed health (χ^2^ (1) = 10.50; Phi = 0.13; *p* = 0.001) in the sense that, as shown in Table 4, individuals who perceived that they had suffered from discrimination gave lower scores for their health. There was also a significant relationship between having experienced perceived discrimination and housing situation (χ^2^ (1) = 10.96; Phi = 0.13; *p* = 0.001): people in literally homeless situations more frequently reported having experienced discrimination (Table 4).

Taking into account the relationship between having suffered from violence while homeless and the other variables, there is a significant relationship between sexual assault and the variables of sex (χ^2^ (1) = 57.01; Phi = 0.30; *p* < 0.001), age (assuming an equality of variances; Levene test: F = 0.3.68; *p* = 0.55; t (600) = 3.12; *p* < 0.001; d = 0.63), housing situation (χ^2^ (1) = 11.75; Phi = 0.14; *p* < 0.001), and nationality (χ^2^ (3) = 18.35; Phi = 0.17; *p* < 0.001). As set out in Table 5, sexual assault was more frequently reported among women aged between 36 and 51 years, of Latin American origin, and in a literally homeless situation.

There are also significant associations between physical violence and self-assessed health (χ^2^ (1) = 6.38; Phi = 0.10; *p* = 0.012), sex (χ^2^ (1) = 24.39; Phi = 0.20; *p* < 0.001), and nationality (χ^2^ (3) = 10.94; Phi = 0.14; *p* = 0.012). In this regard, women, participants of European origin, and participants with lower self-assessed health scores more commonly reported having been victims of physical assaults.

Finally, in terms of verbal assaults, a statistically significant relationship was found with the variables of self-assessed health (χ^2^ (1) = 4.91; Phi = 0.10; *p* = 0.027) and sex (χ^2^ (1) = 18.57; Phi = 0.17; *p* < 0.001). Again, this type of violence appeared more frequently among women and people who provided a negative self-assessment of their state of health.

### 3.2. Logistic Regression Analyses

As set out in Table 6, four logistic regression models were designed with the aim of analyzing the relationship between experiences of discrimination and violence and mental health among PSH. In model 1, being female (OR = 2.234; *p* < 0.001; 95% IC 1.43–3.50), age (OR = 0.970; *p* < 0.001; 95% IC 0.96–0.99) and having a European nationality (not including Spain) (OR = 0.302; *p* < 0.001; 95% IC 0.16–0.57) appear as explanatory categories for impaired mental health among PSH. Being of European origin reduced the likelihood of having impaired mental health. On the contrary, being female increased that likelihood, as did being younger.

This effect is maintained in model 2, where self-assessed health is added to the explanation of the mental health status provided by the sociodemographic variables, with a negative self-assessment of one’s state of health contributing to explaining psychological impairment among participants (OR = 0.262; *p* = < 0.001; 95% IC 0.12–0.57). In other words, PSH who provided more positive assessments of their state of health were less likely to suffer from impaired mental health.

Model 3 incorporates perceived discrimination. As hypothesized, discrimination is an explanatory variable for mental health (OR = 0.458; *p* = < 0.001; 95% IC 0.31–0.68). Specifically, the results for this model show an increased likelihood of suffering from psychological impairment for people who reported having felt that they had suffered from discrimination due to their situation of homelessness.

Finally, model 4 shows that no types of violence (physical, verbal, or sexual) had an impact on mental health among PSH.

## 4. Discussion

The aim of this study was to analyze the impact of experiences of discrimination and violence on mental health among PSH. Table 7 shows the study hypotheses and the conclusions in this respect.

The findings confirm hypothesis 1, which proposed that PSH suffer from high levels of mental health impairment. These findings do not differ from previously published research [56,57,58]. With regard to hypothesis 2, relationships were confirmed between impaired mental health among PSH and certain sociodemographic variables. In this regard, being female [59,60], being young [61], and being an immigrant [62] were associated with higher levels of mental health impairment.

The findings with respect to mental health among women show that female homelessness is characterized by mental health issues. One of the more common explanations for this is that women experience more frequent and severe stressful life events than men in situations of homelessness [63]. Along the same lines, and in relation to age, one potential explanation for the higher levels of mental health impairment among younger people is that they face more risks, traumas, and situations of everyday stress and at the same time have fewer strategies and resources with which to cope with those situations [64], including facing greater barriers to accessing mental health services [65]. 

With respect to the role played by nationality in explaining mental health impairment, the study findings show that mental health is significantly worse among PSH of Latin American origin. In contrast, being of European nationality (not including Spain) is a protective factor in mental health terms. This could be interpreted as resulting from different administrative circumstances and migratory statuses—in other words, the recognition of a regular and stable administrative situation for Europeans compared to immigrants from outside Europe. This differentiation of migratory status is critical for access to rights and, hence, to services, benefits, and resources (including for mental health), which are indispensable in alleviating the psychological impact of social inequalities [66,67]. In this regard, the profile of immigrants in situations of homelessness raises questions concerning the etiology of mental health problems and how they are addressed with people from different cultures [26].

Despite the high frequency of mental health impairment among PSH, participants generally provided a positive assessment of their state of health. This is interesting given the numerous studies confirming that PSH present higher incidences of health problems and illnesses than the population not affected by social exclusion [21]. This apparent contradiction between the perceived general state of health and mental health impairment could be explained by interpreting the responses of PSH as reflecting the positive adjustments they have to make to their self-assessments given their circumstances, ensuring that they maintain high levels of subjective wellbeing despite their objective state of health [68]. 

The findings confirm the third hypothesis—that homelessness creates situations of risk linked to violence and discrimination. The victimization rate found in this study (44.8%) is lower than that in previous research [33,34,69] but in line with the most recent estimates indicating that violence affects between 27% and 52% of PSH [39]. 

The relationship found between mental health and discrimination confirms the fourth hypothesis: there is an association between perceived discrimination suffered by PSH and their mental health [57,70]. In this regard, discrimination was found to be present in the lives of PSH and to contribute to defining their mental health impairment. Discrimination is hence a key factor in the persistence of situations of exclusion in view of the obstacles that discriminatory actions and attitudes pose to accessing stable housing, employment, training, and adequate mechanisms for coping with difficulties [71]. In short, discrimination denies PSH access to the resources, opportunities, and services they need to improve their circumstances, limiting various dimensions of their wellbeing and quality of life, including mental health.

The data obtained do not permit the confirmation of hypothesis 5, which proposed that experiences of violence are associated with impaired mental health among PSH. This may be due to various reasons. First, in line with the contributions of Da Silva Rosa y Passarella Brêtas [72], PSH may have become used to being victims of violence and, moreover, may not perceive certain situations as involving violence. In this regard, PSH may be so used to experiencing mistreatment as part of their homelessness that assaults do not affect their wellbeing or, therefore, their state of mental health. The findings are also likely to be associated with experiences that are difficult to express or share, involving the repression of emotions of guilt and shame [73].

However, there were certain striking findings concerning experiences of violence. In line with previous research [40,74], being female was found to be a fundamental element of vulnerability and risk in terms of violence [33]. Women more frequently report having suffered from all of the kinds of violence subject to analysis, with a notably high risk of sexual assault [75,76,77,78]. This kind of violence was non-existent among the male participants in this study, but almost 14% of women reported having been the victim of an offence. Women aged between 36 and 50 years and who were of foreign origin, mainly from Latin America, faced a particularly high risk of sexual assault. Origin and age are hence risk factors for violence among PSH, particularly among women [79,80]. These findings indicate the importance of specifically exploring how violence affects women in situations of homelessness, particularly its role in defining their processes of social exclusion. Along the same lines, the findings related to nationality indicate that migratory issues are fundamental in analyzing the processes affecting women in situations of homelessness [81].

Finally, the housing situation of participants was another significant variable with respect to violence and discrimination, although not from the perspective of its explanatory power in relation to mental health. It is striking that people in literally homeless situations reported higher levels of violence. One possible explanation is that people in these situations are more frequently exposed in public spaces and hence have less access to protected spaces [82,83]. However, this risk is also associated with social conceptions regarding PSH, particularly with them being seen as people who are less deserving of respect, protection, and proper treatment [84]. It is therefore notable that this risk associated with one’s housing situation does not have an impact on mental health among PSH. It would be interesting for future research to investigate this issue further.

## 5. Conclusions

Many studies have attempted to analyze mental health among PSH and its association with social factors. However, the risk that PSH face in terms of discrimination and violence and their potential repercussions for mental health has barely been examined in Spain. The findings of this research contribute to the generation of knowledge in this respect, showing the presence of two significant issues in participants’ homelessness pathways: their high levels of psychological impairment and their high exposure to violence and discrimination.

The relationship between perceived discrimination and impaired mental health among PSH is also confirmed. However, the findings with respect to the absence of a relationship between mental health and violence, though unexpected, open up potential further lines of research that require exploration, particularly when other dimensions such as one’s perceived state of health affect the impact of social factors on the health of PSH. An in-depth examination of this relationship also requires the consideration of the specific impact that certain variables have on the impairment of mental health among PSH when experiences of violence are involved. In any case, the findings in this respect show that, although discrimination and violence are present in the life pathways of PSH, there are realities that need to be considered in terms of their different conceptualization and their different impact on the mental health of this population. 

This study also shows that, as in the case of the general population, the impairment of mental health among PSH is affected by various elements of inequality that give rise to specific intersections of vulnerability: there are social factors that affect PSH by defining their life pathways but also their mental health. Certain higher-risk factors can be identified: origin, age, sex, and housing situation are important elements for analyzing the social inequalities that define the life pathways of mental health, violence, and discrimination that affect PSH. There is hence a need for continued studies and research specifically analyzing how certain realities affect the mental health of PSH from intersectional perspectives. This entails examining the accumulation of situations involving a disadvantage (being female, an immigrant, and a victim of sexual violence and having impaired mental health), the interaction among those disadvantages, and their effects on PSH.

The study findings have significant implications for policies and practices for supporting PSH in the context of mental health. There is a need to replace fragmented care services with a consolidated new model of comprehensive care that takes into account the social determinants of health and the discrimination and violence that PSH face. In line with the contributions of Bennet-et al. [85], the conclusion is that, although homelessness should be approached as a housing problem, it is also necessary to implement specific measures that consider the health inequalities affecting PSH. Mental health has been repeatedly identified as an important problem for PSH, and action is hence needed to establish community links between mental health services and the current providers of support to PSH. This would imply moving beyond a definition of homelessness that solely focuses on the difficulties of accessing stable and adequate housing.

Although the study findings contribute to understanding the reality of experiences of homelessness and its association with mental health, violence, and discrimination, certain limitations should be taken into account. First, there is a scarcity of literature regarding the relationship between violence, homelessness, and mental health. This makes it difficult to compare the findings of this study with those of similar studies; this should be considered in future research and intervention approaches. Second, although the instrument used to assess mental health has been widely validated, it is not a diagnostic instrument, so epidemiological data have not been obtained. 

In addition, although violence and discrimination have been examined using the same questions that are included in the Spanish Survey on Homelessness, this may represent a limitation given the limited number of questions that this survey contains regarding these variables. It would hence be useful to develop more exhaustive surveys as well as to combine other methodologies (such as narratives) to explore the impact of violence and discrimination on PSH in more detail.

Another limitation is that the study did not take other relevant variables into account. It would be useful for future research to examine how variables including education, marital status, income, social support, social relationships, substance use, and physical illness might direct impact mental health. Above all, it would be worth analyzing how these variables mediate the relationship between mental health, discrimination, and violence.

Finally, the sample was broad but limited in geographical scope (being restricted to the Community of Madrid, Spain). This makes it difficult to generalize the results for PSH at both national and international levels. It would be useful, in this respect, to repeat this study in other Spanish autonomous communities and in different countries, with the aim of comparing the results obtained.

## Figures and Tables

**Table 1 ijerph-20-02034-t001:** Description of the sample.

Variables	Categories	*n*	%
Sex	Male	395	65.5
	Female	205	34.0
Age groups	35 or less	174	28.9
	36–50	147	24.4
	51 or more	281	46.6
Nationality	Spanish	227	37.6
	Other European country	57	9.5
	African	130	21.6
	Latin American	189	31.3
Housing situation	Literally homeless	282	46.8
	Broadly homeless	318	52.7

**Table 2 ijerph-20-02034-t002:** Descriptive analyses for study variables.

Variables	Categories	*n*	%
GHQ-12	Case	429	66.9
	No Case	174	27.1
Self-rated health	Good	527	87.4
	Bad	76	12.6
Perceived discrimination	Never suffered discrimination	311	51.6
	I have suffered discrimination	292	48.4
Violence suffered	Never suffered violence	333	55.2
	I have suffered violence	270	44.8
Type of violence suffered ^1^	Physical violence	208	34.5
	Sexual violence	28	4.6
	Verbal violence	159	26.4

^1^ Multiple-choice question.

**Table 3 ijerph-20-02034-t003:** Relations found between mental health and the independent variables.

Variables	Categories	GHQ-12				Total	
		Case		No Case			
		*n*	%	*n*	%	*n*	%
Sex ***	Male	262	66.3	133	33.7	395	100
	Female	165	80.5	40	19.5	205	100
Age groups **	35 or less	135	77.6	39	22.4	174	100
	36–50	112	76.2	35	23.8	147	100
	51 or more	181	64.4	100	35.6	281	100
Nationality *	Spanish	164	72.2	63	27.8	227	100
	Other European country	29	50.9	28	49.1	57	100
	African	95	73.1	35	26.9	130	100
	Latin American	141	74.6	48	25.4	189	100
Housing situation	Literally homeless	207	73.4	75	26.6	282	100
	Broadly homeless	221	69.5	97	30.5	318	100
Self-rated health ***	Good	361	68.5	166	31.5	527	100
	Bad	68	89.5	8	10.5	76	100
Perceived discrimination ***	Never suffered discrimination	199	64.0	112	36.0	311	100
	I have suffered discrimination	230	78.8	62	21.2	292	100
Type of violence suffered	Physical violence	155	74.5	53	25.5	208	100
	Sexual violence	23	82.1	5	17.9	28	100
	Verbal violence	123	77.4	36	22.6	159	100

* *p* ≤ 0.05; ** *p* ≤ 0.01; *** *p* ≤ 0.001.

**Table 4 ijerph-20-02034-t004:** Relations between perceived discrimination and the control variables.

Variables	Categories	Discrimination				Total	
		Yes		No			
		*n*	%	*n*	%	*n*	%
Sex	Men	191	48.4	204	51.6	395	100
	Women	101	49.3	104	50.7	205	100
Age groups	35 or less	83	47.7	91	52.3	174	100
	36–50	67	45.6	80	54.4	147	100
	51 or more	142	50.5	139	49.5	281	100
Nationality	Spanish	113	49.8	114	50.2	227	100
	Other European country	29	50.9	28	49.1	57	100
	African	60	46.2	70	53.8	130	100
	Latin American	90	47.6	99	52.4	189	100
Housing situation **	Literally homeless	157	55.7	125	44.3	282	100
	Broadly homeless	134	42.1	184	57.9	318	100
Self-rated health **	Good	242	45.9	285	54.1	527	100
	Bad	50	65.8	26	34.2	76	100

** *p* ≤ 0.01.

**Table 5 ijerph-20-02034-t005:** Relations found between violence and the study variables.

Variables	Categories	Type of Violence					
		Physical		Sexual		Verbal	
		*n*	%	*n*	%	*n*	%
Sex	Male	109	27.6 ***	0	0 ***	82	20.8 ***
	Female	98	47.8 ***	28	13.7 ***	76	37.1 ***
Age groups	35 or less	63	36.2	12	6.9 ***	46	26.4
	36–50	58	39.5	12	8.2 ***	50	34.0
	51 or more	86	30.6	4	1.4 ***	62	22.1
Nationality	Spanish	86	37.9 *	5	2.2 ***	57	25.1
	Other European country	25	43.9 *	4	7.0 ***	15	26.3
	African	30	23.1 *	1	0.8 ***	27	20.8
	Latin American	67	35.4 *	18	9.5 ***	60	31.7
Housing situation	Literally homeless	105	37.2	22	7.8 ***	80	28.4
	Broadly homeless	103	32.4	6	1.9 ***	79	28.8
Self-rated health	Good	172	32.6 *	21	4.0	131	24.9 *
	Bad	36	47.4 *	7	9.2	28	36.8 *

* *p* ≤ 0.05; *** *p* ≤ 0.001.

**Table 6 ijerph-20-02034-t006:** Summary of logistic regression analyses.

Variables	Model 1 (Demographic Variables)		Model 2 (Self-Rated Health)		Model 3 (Discrimination)		Model 4 (Violence)	
	B	Exp(B)	B	Exp(B)	B	Exp(B)	B	Exp(B)
Sex (Ref: man)–women	0.804 (0.229) ***	2.234	0.728 (0.231) **	2.072	0.759 (0.234) **	2.137	0.824 (0.248) ***	2.279
Age	−0.030 (0.008) ***	0.970	−0.032 (0.008) ***	0.969	−0.034 (0.008) ***	0.966	−0.035 (0.008) ***	0.965
Citizenship (Ref: Spanish)								
Other European country	−1.198 (0.324) ***	0.302	−1.175 (0.329) ***	0.309	−1.248 (0.337) ***	0.287	−1.241 (0.337) ***	0.289
African	−0.501 (0.299)	0.606	−0.423 (0.300)	0.655	−0.475 (0.306)	0.622	−0.513 (0.310)	0.599
Latin American	−0.469 (0.254)	0.626	−0.405 (0.256)	0.667	−0.438 (0.261)	0.645	−0.457 (0.264)	0.633
Housing (Ref: broadly homeless)—literally homeless	0.198 (0.008)	0.821	−0.189 (0.195)	0.828	−0.106 (0.199)	0.899	−0.124 (0.200)	0.884
Bad self-rated health (Ref: good self-rated health)			−1.340 (0.397) ***	0.262	−1.231 (0.402) **	0.262	−1.253 (0.403) **	0.286
Perceived discrimination suffered (Ref: not suffered)					−0.780 (0.200) ***	0.458	−0.803 (0.208) ***	0.448
Physical violence suffered (Ref: not suffered)							0.168 (0.237)	1.183
Sexual violence suffered (Ref: not suffered)							0.322 (0.576)	1.379
Verbal violence suffered (Ref: not suffered)							−0.050 (0.266)	0.565
Constant	2.579 (0.463) ***	13.183	3.845 (0.606) ***	46.767	4.289 (0.632) ***	72.857	3.992 (0.798) ***	54.176
-2 LL	671.210 ***		656.528 ***		640.770 ***		639.905 ***	
R^2^ (Nagelkerke)	0.102		0.134		0.168		0.170	

** *p* ≤ 0.01; *** *p* ≤ 0.001.

**Table 7 ijerph-20-02034-t007:** Summary of study hypotheses.

Hypotheses	Relevant Results	Confirmation
Hypothesis 1	Nearly 67% of PSH suffer from mental health impairment	Confirmed
Hypothesis 2	Women, young people, and immigrants present higher levels of mental health impairment	Confirmed
Hypothesis 3	More than 48% of PSH have experienced discrimination since becoming homeless, and almost 44% have been victims of violence	Confirmed
Hypothesis 4	There is a greater likelihood of mental health problems for those who have perceived discrimination since becoming homeless	Confirmed
Hypothesis 5	Physical, verbal, or sexual violence did not have an impact on mental health among PSH	Not confirmed

## Data Availability

Not applicable.

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
