# Peer review of "Mental Health and Homelessness in the Community of Madrid (Spain): The Impact of Discrimination and Violence"

_ijerph, 2023, doi:10.3390/ijerph20032034_

Round 1
Reviewer 1 Report
The main objective of this study is to analyze the impact of experiences of violence and discrimination on mental health among people in situations of homelessness. The topic is interesting, however, need some improvement.
1. End of introduction should be explained research gaps based on previous studies.
2. From the beginning I can answer the research questions, so please bring more evidence why we need such research?
3. Background of the study related to research hypotheses are not comprehensive.
4. Please determine a subsection background of the study and research hypothesis.
5. How did authors measure sample size?
6. Table 1 total number of GHQ-12, Violence suffered, and Type of violence suffered are not equal 641.
7. There are some latent variables, why authors didn’t use Structural Equation Modeling (SEM), or PLS?
8. It’s better consider a Table to show the results of their hypotheses.
Reviewer 2 Report
Thank you for the opportunity to review this interesting research titled “Mental health and homelessness: the impact of discrimination and violence”. Data were collected from 641 participants in homelessness conditions in order to analyze the impact of violence and discrimination on their mental health. Overall, the study is organized and scientifically sound.
Some comments to strengthen the work:
1- Contextual information is much needed. This study was conducted in one city in Spain. It is very important for readers and researchers to understand the context of this city and characteristics of its population. This can also help us gauge the generalizability of the findings.
2- Several studies that examined the profile of PSH in Spain can be easily located but none were cited. This makes the significance of current investigation questionable, or at least not well clarified. In other words, what is the added value of this study and how it differs from available ones.
3- The hypotheses did not seem to add anything to the study. They seem to be well-known findings. Further, authors included hypothesis 5 then argued against its validity in the discussion.
4- Lines 98-105 would better be suited under the results section.
5- Was the data collection conducted in a form of Q & A format? Or paper and pencil survey? Either way, details on the process of filling the questionnaire is needed (the seating, were they allowed to ask questions, any incentives?). What about participants who were not able to read or write? Participants with impaired cognition?
6- Analyzing two major variables in the study by using only one question for each is a limitation that needs a rational and acknowledgement. Another major limitation includes the lack of many significant demographic and health-related information that can potentially impact participants mental health.
7- Include psychometrics of the utilized scales for the current sample.
Thank you !
Reviewer 3 Report
The paper investigates the critical topic and public health issue of experiences of violence and discrimination on mental health among people facing homelessness. Findings based on various survey metrics deployed among 641 homeless people provide various values insight, including the association of mental health and impairment and experiences of discrimination, the impact of gender, age, and nationality on psychological impairment, etc. I have a few concerns that I would like the authors to address to improve the quality of the manuscript.
I would recommend that the authors change the title to reflect these findings from Spain or something along the lines of “A community-based study”. I mention this because the findings are specific to a community (Madrid), so generalizations should be made cautiously.
Abstract
● Line 15 “applying” is not the right choice of word. Suggest using “deploying”.
● For findings such as “Perceived experiences of discrimination are associated with higher levels of mental health impairment”, Report the statistical measure as well (p-value/odds ratio/risk ratio etc.)
● Line 23 “This study shows the reality of PSH experiences…” Strong choice of word “reality”. Would suggest toning down.
Introduction comments
● Remove the sentence “PSH are one of the …… discrimination”. Line 51. Repetitive.
● Authors report twice that “age, gender, nationality, and housing situation, as well as their intersectional impact, play a fundamental role” (Line 66 and 70) and cite different studies (which is good). But does not mention how it impacts, for e.g. younger, more impacted, female more impacted, etc….
● Also, for the 5 hypotheses, authors already mention in the literature that PSH people experience higher psychological impairment, age, gender has an impact, etc. So please clarify what specific gaps this study addresses that have not been investigated before. If there’s nothing novel, clarify this is a validation study (which is not the case from my understanding). I think there are a lot of novel components in this study.
Materials and Methods comments
● Sample and Participants: Please add a table with descriptives for the second paragraph.
● Procedure: Please report if the participants were compensated for their time.
● Variables and measures: Line 124 authors mention, “It is frequently used to analyze homelessness [30, 31, 32]” I thought GHQ-12 is used to measure psychological morbidity and mental health problems, not homelessness. Please clarify.
Results
· Line 210, since you had an unequal sample size, I wanted to double-check check you tested for equality of variance before making the assumption. Please confirm.
The rest of the results, discussions, and conclusions looked good.
Round 2
Reviewer 1 Report
The authors amended all of my comments. From my side, the paper already has enough quality to accept for publication. Thank you!
Reviewer 2 Report
Thank you for the opportunity to review the revised version of this manuscript, which seems significantly strengthened. Authors have also made great job contextualizing the study and linking it to the journal's interesting special issue on social Inequalities in Health.